# Comparison of Untargeted and Markers Analysis of Volatile Organic Compounds with SIFT-MS and SPME-GC-MS to Assess Tea Traceability

**DOI:** 10.3390/foods13243996

**Published:** 2024-12-11

**Authors:** Marine Reyrolle, Valérie Desauziers, Thierry Pigot, Lydia Gautier, Mickael Le Bechec

**Affiliations:** 1IPREM, Institut des Sciences Analytiques et de Physicochimie Pour L’environnement et les Matériaux, UMR 5254, Universite de Pau et des Pays de l’Adour, E2S UPPA, CNRS, IMT Mines Ales, Helioparc, 2 Avenue President Angot, 64053 Pau, CEDEX 9, France; 2T Edition, 63 rue Vercingétorix, 75014 Paris, France

**Keywords:** tea, traceability, volatile organic compounds, SIFT-MS, SPME-GC-MS, volatile profiles, untargeted analysis, volatolomics

## Abstract

Tea is one of the most consumed beverages in the world and presents a great aromatic diversity depending on the origin of the production and the transformation process. Volatile organic compounds (VOCs) greatly contribute to the sensory perception of tea and are excellent markers for traceability and quality. In this work, we analyzed the volatile organic compounds (VOCs) emitted by twenty-six perfectly traced samples of tea with two analytical techniques and two data treatment strategies. First, we performed headspace solid-phase microextraction gas chromatography–mass spectrometry (HS-SPME-GC-MS) as the most widely used reference method for sanitary and quality controls of food. Next, we analyzed the samples with selected-ion flow-tube mass spectrometry (SIFT-MS), an emerging method for direct analysis of food products and aroma. We compared the performances of both techniques to trace the origin and the transformation processes. We selected the forty-eight most relevant markers with HS-SPME-GC-MS and evaluated their concentrations with a flame ionization detector (FID) on the same instrument. This set of markers permitted separation of the origins of samples but did not allow the samples to be differentiated based on the color. The same set of markers was measured with SIFT-MS instrument without success for either origin separation or color differentiation. Finally, a post-processing treatment of raw data signals with an untargeted approach was applied to the GC-MS and SIFT-MS dataset. This strategy allowed a good discrimination of origin and color with both instruments. Advantages and drawbacks of volatile profiles with both instruments were discussed for the traceability and quality assessment of food.

## 1. Introduction

Tea, produced from buds and leaves of *Camellia sinensis* plants, is one the most consumed beverages in the world, owing to the diversity of its flavors as well as its health benefits. Historically originating from China, tea is now produced in a large and diverse number of countries throughout Asia, Africa, and the Americas [1]. In 2022, the global production of tea reached 6.7 million tons with China as the leading producer, contributing to 50% of the world output, followed by India (20%) and the two main exporting nations, Kenya (8%) and Sri Lanka (4%) [2]. The soil, the climate, the cultivar selection, the associated plants, and the finesse of the plucking play crucial roles in the development of the aromatic potential of tea [3,4]. Moreover, numerous transformation processes of tea leaves have led to the creation of several tea grades [5,6] with different oxidation, fermentation, and roasting rates that result in several tea colors (green, oolong or blue-green, black, dark, red, white, and yellow). According to the FAO, the global tea trade market represents more than USD 9.5 billon with a median price to USD 2.4 per kg [7]. To protect this market, several tea-growing regions developed their labels, for example, Protected Designation of Origin (PDO) Darjeeling tea from India, Protected Geographical Indication (PGI) for Ceylon tea (Sri Lanka), and Geographical Indication (GI) for Kenya or Oolong tea from Taiwan. The aim of these labels is to guarantee consumers not only the geographical origin of the tea, but also a certain expected quality that justifies its price. However, as for many high value products, the tea industry faces several types of fraud: mislabeling and false advertising, substitution of ingredients, adulteration and counterfeit. Thus, tea fraud affects both the economy, by pushing down prices of legitimate products, undermining the livelihood of producers, and public health. For consumers, it can lead to consumption of substandard or even harmful products associated with risks of allergic reactions or long-term health complications from ingesting chemicals or toxins [8]. The overall structure of the tea chain varies from country to country but is generally complex, involving numerous small producers, cooperatives, and tea factories, with several stages and a wide range of actors, from production to consumption, which increases the risk of admixture and counterfeit.

Sanitary controls and the certification of labels have, up to now, been the main tools to reinforce the traceability and authenticity and to restore consumer confidence. Food traceability includes all the declarative information regarding the geographical course of the product, from its production to its consumption. Food authentication, on the other hand, is the global process of verifying the conformity of a product with its label description, including origin information (species, geographical area, etc.), agricultural practices (conventional, organic agriculture, etc.), and processing methods.

For the assessment of food authenticity and traceability, control agencies rely on analytical techniques such as separation techniques coupled with mass spectrometry (LC/MS, GC/MS), near-infrared spectrometry (NIRS), and isotope ratio mass spectrometry (IRMS) that have been developed in recent years [9].

In the food industry, the volatile organic compounds (VOCs) are essential due to their role in the taste and flavor perception. The volatile profile of a food product can be affected by geographical origin, seasonal variation, manufacturing processes, and storage conditions. This profile includes a large number of different compounds, sometimes hundreds of varying chemical classes, that can be present at very different concentrations. The identification and quantification of VOCs has always been a challenge due to their diversity and the complexity of food matrices. Headspace (HS) solid-phase microextraction (SPME) coupled with gas chromatography–mass spectrometry (GC-MS) is one of the most commonly used methods for the analysis of VOCs in food matrices [10,11,12]. This technique allows (1) separation of compounds based on their physicochemical properties and their affinity for the capillary column, (2) identification thanks to their fragmentation profile with electronic impact mass spectrometry, and (3) their quantification with calibrations. The quantification of dozens of volatile markers has been used for VOC analysis of tea to study different steeping temperatures [13], varieties of cultivars [12,14], fermentation processes [10,15], and geographical origins [16,17].

In recent years, direct-injection mass spectrometry (DIMS) instruments such as proton-transfer-reaction mass spectrometers (PTR-MS) or selected-ion flow-tube mass spectrometers (SIFT-MS) have been developed for rapid, non-invasive, and direct on-line measurement of VOCs [18,19,20,21]. The selectivity of these technologies is not based on the separation of analytes but on soft chemical ionization of the VOCs emitted by products. These instruments are becoming increasingly popular due to their high sensitivity and speed [22]. In the food industry, the SIFT-MS technique has been used as a tool to authenticate olive oils of Mediterranean origin [23] and their adulteration [24], for geographic traceability of Moroccan argan oils [25,26], to discriminate the volatolome from *Vitis vinifera* berries [27], and to discriminate cheeses [28,29]. Only a few studies to date have compared the performances of DIMS and GC-MS for VOC monitoring [30,31,32].

In this work, VOCs analysis with HS-SPME-GC-MS and HS-SIFT-MS was performed on 26 tea samples of perfectly certified origins, collected directly from producers in order to assess the sample origin and quality and avoid mixed samples. We evaluated two post-treatment strategies of the GC-MS and SIFT-MS results: targeted analysis based on the monitoring of a list of markers and untargeted analysis based on profiles or fingerprints. The main point of this study is to show that it is possible to re-analyze results of analyses already recorded with a new strategy to obtain complementary and/or more effective results.

## 2. Materials and Methods

### 2.1. Samples

A tea expert collected 26 tea samples with certified origins directly from the factory. The 26 teas came from seven different countries: Nepal (*n* = 3), China (*n* = 9), Vietnam (*n* = 5), India (*n* = 3), China_Taiwan (*n* = 2), Sri Lanka (*n* = 2), and Japan (*n* = 2), amounting to a total of twelve different regions, and they had five different colors: green (*n* = 6), black (*n* = 10), dark (*n* = 3), white (*n* = 4), and oolong (*n* = 3) (Table 1).

### 2.2. Headspace–Solid-Phase Microextraction–Mass Spectrometry–Flame-Ionization Detector (HS-SPME-GC-MS-FID) Analysis

One gram of dry tea was placed into a 10 mL headspace vial and immediately sealed. After a headspace equilibration phase at 60 °C for 10 min, VOCs were extracted and pre-concentrated with an SPME fiber (30/50 µm PDMS/DVB/CAR Supelco, St. Louis, MO, USA) to extract a maximum number of compounds with a wide range of physicochemical properties. The SPME fiber was exposed to the tea sample headspace for 12 min at 60 °C and desorbed for 5 min at 250 °C into a GC-MS/FID injector.

A gas chromatograph (Agilent 7890B, Agilent, Santa Clara, CA, USA) coupled with an MS detector (Agilent 5977B) and a flame-ionization detector and connected to a multifunction autosampler (Combi-Pal, CTC Analytics, Zwingen, Switzerland) was used to analyze the volatile compounds in the tea samples. Separation of the volatile compounds was achieved with an ELITE-5MS capillary column ((5%-phenyl)-methylpolysiloxane, 30 m × 250 μm, 0.25 µm film thickness, PerkinElmer, Waltham, MA, USA). Helium 6.0 (Air Liquide, Paris, France) was used as the carrier gas, with a constant flow rate of 1 mL min^−1^. The injection port was equipped with a 0.8 mm internal diameter liner and maintained at 250 °C in splitless mode. The oven temperature was initially held at 40 °C for 4 min, followed by a first temperature rise to 90 °C at a rate of 15 °C/min, held at 250 °C for 4 min, and a second temperature rise to 250 °C at a rate of 10 °C/min, and then held at 250 °C for 5 min. The total analysis time was 32 min 33 s. The mass spectrometer was operated using electron impact (EI) mode with an electron energy of 70 eV. The ion source, transfer line, and quadrupole temperature were set at 280 °C, 250 °C, and 150 °C, respectively. Acquisition was carried out in scan mode, with the mass ranging from 15 to 250 atomic mass units (amu). A GC-MS chromatogram of a sample of tea is presented in Appendix A in Appendix A. The volatile compounds were tentatively identified using the standard NIST 14 library for mass spectra. Each tea sample was analyzed in triplicate.

### 2.3. Headspace-SIFT-MS

Ten grams of dry tea were added to a 1 L bottle. The bottle was fitted with a polypropylene screw cap with two tight connection ports fitted with 0.6 cm PFA tubing. The first one was connected to the SIFT-MS and the second one to a 1 L Tedlar^®^ bag (Supelco, Bellefonte, PA, USA) filled with zero dry air (ZeroAir Alliance ZA1500, F-DGSi, Evry, France) to compensate for the volume used for the SIFT-MS analysis. The closed bottle was previously incubated for 1 h at 60 °C before performing the positive and negative SIFT-MS full scan analysis.

A SIFT-MS Voice200™ Ultra (SYFT Technologies, Christchurch, New Zealand) equipped with a dual source producing positive and negative soft-ionizing precursor ions (H_3_O^+^, O_2_^●+^, NO^+^, O^●−^, OH^−^, O_2_^●−^, NO_2_^−^, and NO_3_^−^) in a single scan was used. Nitrogen (Air Liquide, Paris, France) was used as the carrier gas, and the sample was introduced through a temperature- (110 °C) and flow-controlled (20 mL min^−1^) sample line (High-Performance Inlet HPI^®^). The instrument was calibrated daily with a standard gas containing standards at 2.0 ppmV in nitrogen (ScottTM gas mixtures, Air Liquid, Plumsteadville, PA, USA). A blank measurement of each empty bottle was performed before introduction of the sample. The scanned raw data files containing product ion intensities within the 15–250 *m*/*z* range were collected.

### 2.4. Data Processing and Statistical Analysis

Hereafter, VOC refers to any volatile organic compound, a marker refers to an identified volatile molecule, and a profile refers to a set of VOCs that evolve in the same trend in a group.

Targeted approach: Two tables were constructed with GC-MS and SIFT-MS results, corresponding to “pseudo-quantification” of markers for each sample (mean of triplicates and samples from same category: region/color). The molecules identified by GC-MS with a NIST database percentage match higher than 70% were retained as markers, and the corresponding peak areas were measured using the FID signal (Appendix A). This list of markers was used to construct a quantification method for SIFT-MS measurements. Thus, the product ions corresponding to these markers were identified in the data software LabSyft^®^ release 1.6.2 (Appendix A). After removing the conflict ions, corresponding to product ions with the same *m*/*z* ratio from different molecules, a Multi-Ion Monitoring method was developed and applied to the full scan measurements to quantify these markers (in ppbV). Principal component analysis (PCA) and partial least square discriminant analysis (PLS-DA) were then conducted on these datasets to sort samples.Untargeted analysis of HS-SPME-GC-MS was performed with ChromCompare+ software version 2.1.4.1 (Markes International, Bridgend, UK). As described by Spadafora et al., a data alignment algorithm was applied to one of the 78 chromatograms [33] to overcome retention-time drift observed across the dataset. The method used the mass spectral data to automatically align each chromatogram with a ‘reference’ chromatogram. No additional data pre-processing was necessary. Secondly, an untargeted tile-based approach was performed on the aligned chromatograms, dividing each chromatogram into 4000 tiles. An initial filtering process was applied with a minimum absolute intensity of 1000 and a minimum label frequency of 100%. A feature discovery algorithm was then applied by selecting a descriptive label (country, region, or color). The 400 most discriminative features were then extracted with the corresponding pic areas before statistical analysis (PCA and PLS-DA) and graphical representations.Untargeted analysis of SIFT-MS analysis: The algorithm developed in our previous work was used to extract the 1888 ions produced by the full scan analysis [29]. The data pre-processing consisted of subtracting the background noise (signal of empty bottle) for every replicate then averaging the triplicates for each sample to obtain a single value of signal intensity (count s^−1^) for each ion. To perform statistical analysis and to highlight differences between samples, the dataset was cleaned by suppressing the quantitative variables, which are constant among all samples (variance equal to zero). R language [34] was used with the “corrplot” package [35] for correlation visualization and the “MixOmics” package [36] to represent the dispersion and discrimination of the samples. Supervised methods (sparse partial least squares–discriminant analysis, PLS-DA) were applied to the datasets. Appendix A in SI presents an example of Heatmap sPLS-DA according to the color of tea samples and the 1880 product ions.

## 3. Results

### 3.1. Targeted Markers Monitoring Strategy

The objective was first to compare the performance of both VOC analytical techniques (HS-SPME-GC-MS and SIFT-MS) for monitoring of markers and to discriminate the tea samples according to their origins or transformation processes. From the set of chromatograms recorded with this tea collection, we looked for compounds that met the following requirements: they had to be well separated from other constituents by the chromatographic method, have a high confidence score compared to NIST data (>70%) for mass spectrometry identification, and present a strong FID signal. We thus selected 48 main compounds expressed in the headspaces of our samples, including 12 alcohols, 11 terpenes, 8 aldehydes, 5 ketones, 3 esters, 4 alkanes, 3 acids, and 2 other compounds (furan, 2-pentyl, and caffeine). Thirty-seven of the 48 molecules were found in the LabSyft database, with 138 corresponding product ions. However, among these product ions, 90 were conflict ions, meaning that two or more molecules can generate an ion with the same *m*/*z* ratio. These signals have to be removed for the quantification of markers. Finally, only 48 product ions were not suspected to be interfered with by another molecule in the list. Hence, 9 compounds could not be quantified (marked with an asterisk in Table 2). PLS-DA analysis was applied on each dataset (GC-FID peak areas and SIFT-MS concentrations) to try to discriminate the two descriptive parameters “Country_region” and “Color”.

#### 3.1.1. Targeted Origin Discrimination

The mean values of triplicates of the FID peak areas of the 48 different markers were calculated for every sample. A PLS-DA was then performed on the dataset according to the descriptive parameter “Country_region” to look for correlations. The results are presented in Figure 1A according to the first and the second components of the PLS-DA. Four different classes of samples were clearly separated: Japan_Kyushu + (circled in green), Nepal_Ilam Valley × (circled in red), Vietnam_Lai Chau ■ (circled in blue), and India_Darjeeling ▲ (circled in orange). A group of eight samples from different origins (China_Yunnan ●, China_Taiwan_Ali Mountain ▽, Sri Lanka_Matara ✧, and Sri Lanka_Ratnapura **✦** (framed with dot lines)) were not separated and exhibited strong similarities according to this test. Interestingly, these tea samples shared the same type of plant cultivar: large-leaf Camellia sinensis. With the second representation, according to the second and the third components of the PLS-DA (Figure 1B), three other origins could be separated (China_Taiwan_Ali Mountain ▽, circled in grey; the two regions of Sri Lanka ✧✦ together, circled in purple; and the three regions of China, circled in black). Only one sample from Vietnam_Lao Cai ■ was not successfully separated from the other samples with this test. These results confirmed that tea samples can be sorted according to their origin with GC-MS peak areas, as previously demonstrated for green teas by Ye et al. 2022 or for black teas by Yun et al. 2021 [16,37]. However, this study showed that this classification can also be achieved with a collection of teas from different regions and different grades (colors). The contributions of the 48 molecules to the classification of the samples according to their origin are shown in Appendix A.

With the same approach, the mean values of triplicates of the HS-SIFT-MS measurements in MIM mode were also calculated for every sample. A PLS-DA was performed on the dataset, but no clear separation according to the region of origin was highlighted, neither with components 1 and 2 nor with components 2 and 3 (Figure 1C,D) of the test. On the contrary, the different samples were spread over the representation, indicating a high level of heterogeneity. This result may be explained by the small number of ions that were used for this statistical test. Indeed, the conflict ions inherent to DIMS technology greatly limit exploitation of these instruments for the quantification of markers in complex matrices. However, Yener et al. succeeded in finding statistical correlations of VOC profiles of green and black teas according to their origin using PTR-ToF-MS [38]. The authors selected 257 mass peaks before applying statistical tools to trace the origin of the green and black teas. With a high-resolution instrument, the authors also succeeded in tentative identification of a number of chemical compounds (61) corresponding to the most discriminant ions.

#### 3.1.2. Targeted Transformation Process Discrimination

The same statistical analysis of the mean values of marker signals with HS-SPME-GC-MS/FID and HS-SIFT-MS was also performed to look for correlations with the descriptive value “color”, reflecting the different transformation processes of teas. The results presented in Figure 2 show high dispersion of the samples with the same color. With the separative technique (Figure 2A), the 3 oolong teas, the 5 green teas, and the 10 black teas tended to separate from each other. Unfortunately, dark and white tea samples were placed in a median position and made segregation of the five grades difficult. No more explanatory information was provided with the third component of the PLS-DA test. Several studies have already tried to characterize the transformation process of tea with markers. Thus, Alasalvar et al. investigated the characterization of different quality grades of black teas from Türkiye using several techniques, including HS-GC-MS [39]. They concluded, regarding the importance of the concentration levels of some of the 57 volatile compounds, that there was no clear distinction between the seven grades. Lin et al. 2013 [14] analyzed 26 VOCs emitted by different varieties of oolong teas using HS-SPME-GC-MS, and they obtained good separations, with a limit in the misclassification of cheaper products. Zhang et al. 2013 analyzed several grades of teas (green, oolong, and black) using two-dimensional GC-ToF-MS after simultaneous distillation extraction, and they succeeded quite well with the separation of sample colors thanks to 3000 detected peaks, of which, 450 were tentatively identified [40]. Finally, 74 differential compounds allowed discrimination between the three grades.

This work with markers was performed with 48 identified molecules. These molecules were chosen on the basis of the most concentrated and best separated by GC, thereby allowing for a good match score with the NIST database. Contrary to Whang et al., we did not succeed in sorting the different sample colors. Nevertheless, the choice of markers was oriented by the nature of the SPME fiber and the phase of the GC column selected. These choices were made based on the literature and the aim to detect the widest range of volatile chemical compounds. Was it the best choice to discriminate the origins or colors of our samples? Clearly not. With GC-MS, and even more so with SIFT-MS SIM mode, the number and the choice of key markers is essential. The emerging approach of untargeted analysis starts with the hypothesis that the key molecules are unknown and not necessarily the most concentrated.

### 3.2. Untargeted Analysis of Volatile Profiles of Teas

The global signals of HS-SPME-GC-MS and HS-SIFT-MS were analyzed to investigate the contribution of the volatile profiles to discriminate tea samples according to the geographical area of production or color. The post-treatment of the entire raw total-ion chromatogram (TIC) trace of the chromatograms with ChromCompare+ included an alignment protocol, a filtering step, and a feature allowing discovery according to the region or the color parameter. The 400 features were then further explored to highlight correlations. For the SIFT-MS full scan measurements, the entire signals for the product ions from a 15 to 250 *m*/*z* ratio for the eight different precursors (1880 ions) were analyzed to discriminate the two descriptive parameters (country_region, and color).

#### 3.2.1. Untargeted Origin Discrimination

The sPLS-DA analysis of volatile profiles obtained by GC-MS allows good discrimination of the geographical areas (Figure 3A), especially for teas from China_Taiwan_Ali Mountain ▽ (circled in grey), Vietnam_Lai Chau ■ (circled in red), and Japan_Kyushu + (circled in green). The other samples were not separated by regions, but three separate clusters could be drawn. The first (square 1) contained the samples from China_Yunnan ●, Vietnam_Lao Cai ■, and the two provinces of Sri Lanka. The proximity of these samples cannot be explained by geographical proximity or their proximity with the nearby group Vietnam_Lai Chau ■. However, the knowledge of the products allowed another reason to be proposed that could explain these proximities. All these teas were obtained from large-leaf cultivars of *Camellia sinensis*, meaning that the cultivar may have a great impact on the volatile profile of teas. This cluster was nevertheless successfully separated according to the third component (Figure 3B). The two Chinese provinces Jiangsu ● and Zhejiang ●, located in the east of China, also presented high statistical proximity of their volatile profiles. The geographical proximity may explain this observation and, moreover, the cultivars used in both provinces are small-leaf teas (square 2). Finally, the teas from India_Darjeeling ▲ had very similar volatile fingerprints to those from Nepal_Ilam Valley **×** and formed a third cluster (square 3). Indeed, these two regions are geographically very close, but the cultivars used for these teas are also very similar. In addition, the cultivars used for these two regions are mixtures of small-leaf teas like those used for China_Jiangsu and China_Zhejiang, so it is consistent to find proximity between these tea samples. The untargeted analysis of HS-SPME-GC-MS allowed tracing of some of the tea origins as for the targeted analysis with 48 VOCs, but this approach also highlighted the statistical proximity of GC-MS fingerprints with the type of cultivars. The elucidation of the tea plant cultivar contribution on VOCs profiles would be confirmed with phylogenic distance evaluation between samples. However, the DNA analysis of oxidized or fermented tea leaves is challenging and requires specific skills.

The results of the sparse PLS-DA analysis from the volatile profiles obtained by SIFT-MS (Figure 3C,D) also showed good discrimination of tea samples from Japan_Kyushu + and Vietnam_Lai Chau ■, as previously shown with untargeted GC-MS. The other discriminated groups were China_Zejiang ●, China_Jiangsu ●, and China_Yunnan ● with the first and the second components of the test. Interestingly, with the representation of components 2 and 3 of the test, the large-leaf Camellia sinensis samples grouped together in a cluster, and the samples from China_Taiwan_Ali Mountain **▽** were separated. However, no clear separation was observed for the India_Darjeeling ▲ samples. Thus, the proximity of small-leaf *Camellia sinensis* samples highlighted using GC-MS was not clear according to the three first components of the sPLS-DA test on SIFT-MS fingerprints.

#### 3.2.2. Untargeted Analysis According to Transformation Processes

Contrary to targeted analysis with the 48 VOCs, the untargeted data post-treatment of GC-MS and SIFT-MS full scan analysis allowed good discrimination of tea colors with components 1, 2, and 3 of the sPLS-DA (Figure 4). Thus, oolong and dark teas were clearly separated with components 1 and 2 of the GC-MS analysis, whereas the three other grades were grouped with components 2 and 3 (Figure 4B). From this supervised analysis of the 400-features dataset, the most contributing variables of the components 1, 2, and 3 were extracted and are presented in Appendix A SI. From these results, it was possible to go back to the chromatograms and look for the involved molecules. Then, boxplots of features’ intensities can be calculated (Appendix A SI). With SIFT-MS analysis, the most different grade of tea was green tea (Figure 4C). The four other grades were separated with components 2 and 3 of the sPLS-DA (Figure 4D). The color of tea is related to the processing of the tea leaves after harvesting [2]. The different stages of processing lead to different levels of oxidation, fermentation, and roasting. Oolong teas are the most oxidized, dark teas are fermented, and green teas are those that undergo the least amount of processing, as they are neither oxidized nor fermented. Black teas are generally produced with very heterogeneous processes depending on the cultures, thus leading to heterogeneous levels of roasting. These transformation processes of tea, which modify their chemical composition, were clearly visible by untargeted analysis with either HS-SPME-GC-MS or SIFT-MS.

## 4. Discussion

The aim of this study was to compare the performances of two VOC analytical techniques (HS-SPME-GC-MS/FID and HS-SIFT-MS) using a marker-monitoring method and a volatile profile method to discriminate tea samples with descriptive variables, e.g., the geographical area of production and the transformation processes. First, our results show that the monitoring of markers with HS-SPME-GC-MS/FID could discriminate the geographical area of production of our samples, highlighting the impact of the cultivars, but did not allow discrimination according to the colors of the teas. However, for this method, the selection of the key compounds was performed once the chromatograms had already been obtained, with adapted protocols of HS-SPME extraction and GC separation of the sample database, thus influencing the choice of compound selection. It appears that the 48 selected compounds were more discriminating in terms of the geographical area and cultivar than for the process (color). Second, with SIFT-MS, no discrimination was obtained with a SIM method established with the same list of molecules, neither for the origin nor the color. Nevertheless, a lot of conflict ions were found with the LabSyft^®^ quantification method and then reduced the number of analyzed signals (Appendix A). Finally, only 28 of the 48 compounds were analyzed, with only 48 non-conflict product ions of the 138 possible ones. SIFT-MS instruments have some well-known drawbacks due to the ion–molecule reaction in soft-chemical ionization [41]. Isomeric compounds can form the same product ions with the same *m*/*z* signal, which prevents their quantification [42]. By knowing the molecules being searched, withdrawal of the conflict ions for the follow-up can be envisaged. On the other hand, in the case of complex matrices such as foodstuffs, unknown molecules present in the matrix can generate conflict ions without them being known. This limits SIFT-MS monitoring for quantification in complex matrices.

Another way to use the data of VOC measurements is untargeted analysis. We applied post-treatments to HS-SPME-GC-MS and HS-SIFT-MS full scan data to look for correlations. We demonstrated that untargeted analysis of GC-MS measurements allowed for the distinction of five geographical origins and grouped several samples that share the same type of cultivars. This suggests that the 400 most discriminant features selected by the ChromCompare + software do not correspond with the information provided by the 48 compounds selected with the targeted approach. Indeed, among these 400 features, only 24 molecules on the list were found. The identification of the other molecules was difficult because of weak correspondence with the NIST Database. This observation may guide scientists regarding the great importance of the markers used for the discrimination. With SIFT-MS, the untargeted analysis permitted correlations to be found that were not observed with the SIM method. It seems clear from these results that the choice of markers is a key element for samples classification based on descriptive variables. The use of an untargeted analysis strategy for GC-MS and SIFT-MS measurements avoids any marker-selection bias and maximizes sample sorting. Five different tea origins were separated with a PLS-DA analysis of the full scan data. The impact of large-leaf tea cultivars was also highlighted by this methodology. As with untargeted GC-MS, it is possible to evaluate the transformation process of teas with untargeted SIFT-MS.

Both analytical methods are relevant for the analysis of VOCs in food matrices. However, they each have their specific advantages and drawbacks. The comparison of these two methods based on the following four parameters is summarized in Table 3.

Speed: Time is an important factor in the food industry, which involves the classification of a lot of samples. HS-SPME-GC-MS/FID analysis does not require a difficult preparation step and can readily be automated. However, it takes several tens of minutes for one sample. HS-SIFT-MS analysis can also readily be automated and does not require a lengthy preparation, although a full scan record takes 3 to 6 min depending on the number of precursor ions. This high-speed instrument allows kinetic studies with high throughput.

Selectivity: The selectivity of HS-SPME-GC-MS/FID is based on the composition of the SPME fiber and the GC column that drives the detected chemical compounds. Separation protocols (time, temperature, etc.) are generally adapted based on a selection of molecules that will also drive the resolution. Mass spectrometry detection allows identification with databases (NIST or other), whereas FID detection allows straightforward and robust quantification. SIFT-MS provides chemical selectivity with precursor ions. Depending on the ions, some chemical families can be addressed by this type of instrument. However, the absence of separation leads to the risk of conflict ions, coming from several analytes that result in ions with the same *m*/*z* ratio. With complex matrices, these conflict ions are inevitable and require elimination of such inexplicable signals. This data-filtering step limits the number of molecules that can be measured at the same time. The unitary mass resolution of the instrument, the soft-chemical ionizations, and these ion conflicts limit the identification of unknown molecules in the matrix.

Sensitivity: SPME is a sampling technique that greatly increases the limit of detection of a given molecule by increasing the time and the temperature of the fiber exposure. However, with complex matrixes, where a lot of different molecules with different concentration ranges are mixed, SPME may become saturated by the most concentrated molecule, resulting in the less concentrated ones not being detected. Nevertheless, the global sensitivity of this technique is in the range of ppbV for most VOCs [5]. The sensitivity of SIFT-MS is driven by the chemical reactivity inside the flow tube and the kinetic constant k provided in the software database. Only the 400 molecules presently recorded in the database can be directly quantified with SIFT-MS. If the molecule of interest is unknown, the database may be updated after calibrations. Most of the VOCs can be detected at a range of ten ppbV with SIFT-MS [41].

Fingerprints: Both technologies can be exploited with untargeted strategies. For GC-MS, several software solutions are available on the market. To our knowledge, only lab-made software solutions are available for SIFT-MS full scan treatment.

## 5. Conclusions

This preliminary work shows the potential of profile/fingerprint VOC analysis to discriminate the origin and transformation processing of tea samples. **First**, it clearly appears that the choice of markers drives the performance of targeted GC-MS analysis. In this study, the 48 chosen molecules allowed discrimination of the origin but not the transformation process. **Second**, quantification of a long list of markers with SIFT-MS in SIM mode leads to a high number of conflict ions, thereby limiting the resolution of the discrimination. **Third**, post-treatment of GC-MS raw data highlights new correlations with the transformation process that were not explained with markers. The choice of markers or features (in the case of ChromCompare + software) was oriented by the composition of the sample set. The untargeted analysis of SIFT-MS full scan data makes it possible to analyze complex matrices with statistical profile comparisons. These analyses based on volatile profiles could be techniques of choice for discriminating products with a label such as a Protected Indication of Origin (PGI), a Traditional Speciality Guaranteed (TSG), or a label defining a superior quality. In the current study, a limited number of samples were used, but the results are promising for further study of a larger number of samples. Separation according to origin and/or color could be greatly improved with a sample collection more suited to the purpose. For example, by focusing on the construction of a sample bank of black teas, discrimination of the geographical origins would probably be achieved more readily. However, knowledge of the sample was also a key parameter that allowed us to understand the clustering according to cultivar proximity. We do hope that this work will encourage teams who carry out traditional VOCs analysis using marker lists to consider taking their raw data and analyzing it with non-targeted tools. Finally, learning models could be envisaged in order to highlight fraud by comparison of volatile profiles.

## Figures and Tables

**Figure 1 foods-13-03996-f001:**
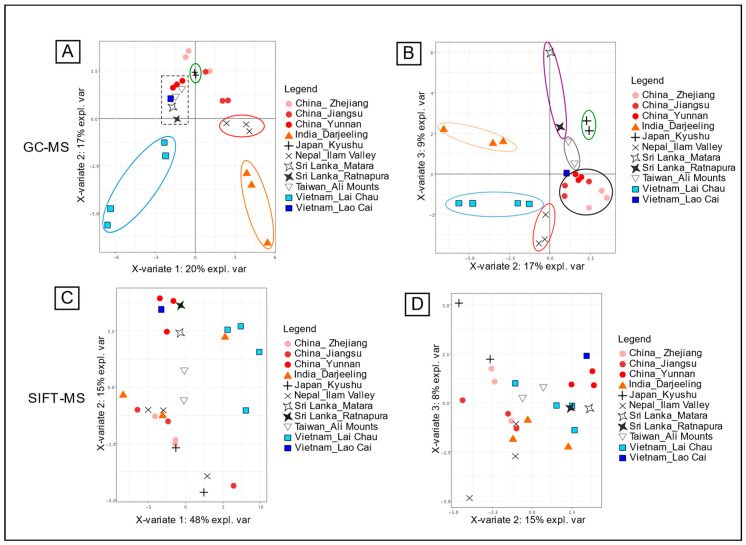
Individual plots of PLS-DA for country discrimination. (**A**) PC1 and PC2 plot for the GC-FID data. (**B**) PC2 and PC3 plot for the GC-FID data. (**C**) PC1 and PC2 plot for the SIFT-MS data. (**D**) PC2 and PC3 plot for the SIFT-MS data.

**Figure 2 foods-13-03996-f002:**
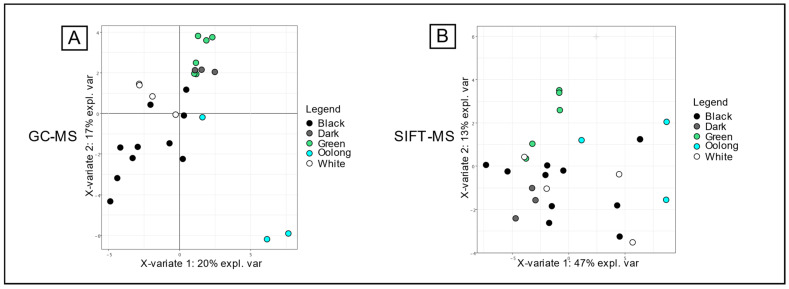
Individual plots of sparse PLS-DA for color discrimination. (**A**) PC1 and PC2 plot for the GC-FID data. (**B**) PC1 and PC2 plot for the SIFT-MS data.

**Figure 3 foods-13-03996-f003:**
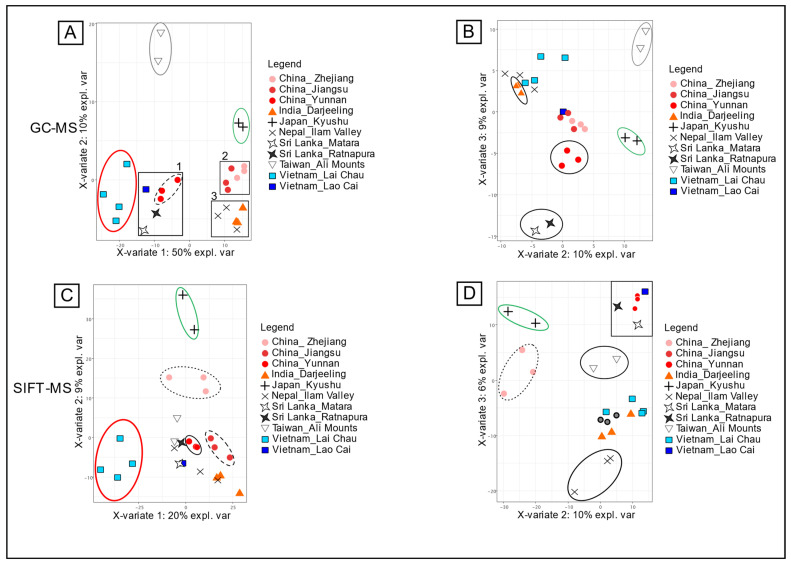
Individual plots of sparse PLS-DA for country discrimination. (**A**) PC1 and PC2 plot for the volatiles profile obtained by GC-MS/FID. (**B**) PC2 and PC3 plot for the volatiles profile obtained by GC-MS/FID. (**C**) PC1 and PC2 plot for the volatile profile obtained by SIFT-MS. (**D**) PC2 and PC3 plot for the volatiles profile obtained by SIFT-MS.

**Figure 4 foods-13-03996-f004:**
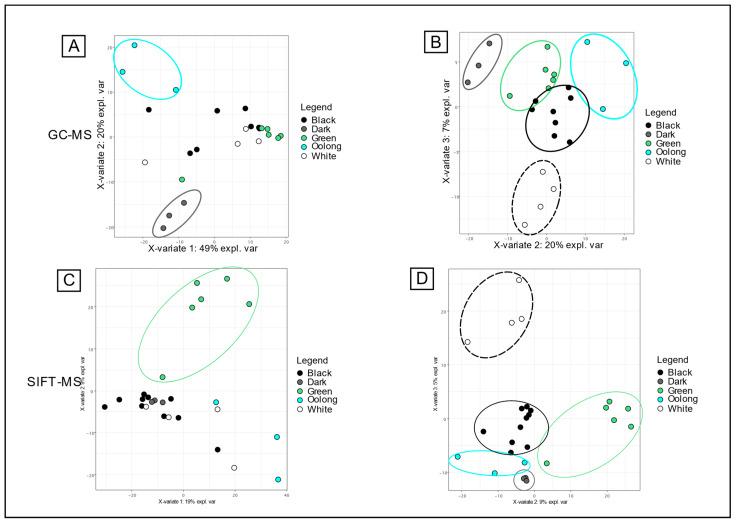
Individual plots of sparse PLS-DA for color discrimination. (**A**) PC1 and PC2 plot for the volatiles profile obtained by GC-MS/FID. (**B**) PC2 and PC3 plot for the volatiles profile obtained by GC-MS/FID. (**C**) PC1 and PC2 plot for the volatiles profile obtained by SIFT-MS. (**D**) PC1 and PC2 plot for the volatiles profile obtained by the SIFT-MS data.

**Table 1 foods-13-03996-t001:** T Description of the tea samples.

Country	Region	Color	Number of Samples
China	Jiangsu	Black	3
China	Yunnan	Dark	3
China	Zhejiang	Green	3
China_Taiwan	Ali Mountain	Black	1
China_Taiwan	Ali Mountain	Oolong	1
India	Darjeeling	Black	3
Japan	Kyushu	Green	2
Nepal	Ilam Valley	White	3
Sri Lanka	Matara	Black	1
Sri Lanka	Ratnapura	Black	1
Vietnam	Lai Chau	Black	1
Vietnam	Lai Chau	Oolong	2
Vietnam	Lai Chau	White	1
Vietnam	Lao Cai	Green	1

**Table 2 foods-13-03996-t002:** Molecules identified by HS-SPME-GC-MS. The crosses indicate the unknown compounds in the LabSyft^®^ database, the check marks indicate the known molecules, and the asterisks indicate the molecules that were excluded because of conflict ions.

Compounds	CAS	MM (g/mol)	Family	SIFT-MS
1-Penten-3-ol	616-25-1	86.13	alcohol	√ *
1-Pentanol	71-41-0	88.15	alcohol	√
2-Penten-1-ol	1576-95-0	86.13	alcohol	√ *
3-Hexen-1-ol	544-12-7	100.16	alcohol	√ *
2-Hexen-1-ol	928-95-0	100.16	alcohol	√
1-Hexanol	111-27-3	102.17	alcohol	√
1-Octen-3-ol	3391-86-4	128.21	alcohol	√ *
Benzyl alcohol	100-51-6	108.14	alcohol	√
Linalool oxide	5989-33-3	170.25	alcohol	x
trans Linalool oxide	34995-77-2	170.25	alcohol	x
Phenylethyl alcohol	60-12-8	122.16	alcohol	√
trans-Linalool 3,7 oxide	39028-58-5	170.25	alcohol	x
beta-Myrcene	123-35-3	136.23	terpene	√
D-Limonene	5989-27-5	136.23	terpene	√ *
beta-cis-Ocimene	3338-55-4	136.23	terpene	√ *
Linalool	78-70-6	154.25	terpene	√
beat-Cyclocitral	432-25-7	152.23	terpene	√
Linalyl acetate	115-95-7	196.29	terpene	√
D-Carvone	2244-16-8	150.22	terpene	√
Geraniol	106-24-1	154.25	terpene	√
alpha-Longipinene	5989-08-2	204.35	terpene	x
beta-Ionone	14901-07-6	192.3	terpene	√
beta-Ionone epoxide	23267-57-4	208.3	terpene	x
Pentanal	110-62-3	86.13	aldehyde	√ *
Hexanal	66-25-1	100.16	aldehyde	√ *
2-Hexenal	6728-26-3	98.14	aldehyde	√
Heptanal	111-71-7	114.19	aldehyde	√
Benzaldehyde	100-52-7	106.12	aldehyde	√
Octanal	124-13-0	128.21	aldehyde	√
2,4-Heptadienal	4313-03-5	110.15	aldehyde	√
Nonanal	124-19-6	142.24	aldehyde	√ *
3-Penten-2-one, 4-methyl	141-79-7	98.14	ketone	x
2-Heptanone	110-43-0	114.19	ketone	√
Butyrolactone	96-48-0	86.09	ketone	√
5-Hepten-2-one, 6-methyl	110-93-0	126.2	ketone	√
3,5-Octadien-2-one	30086-02-3	124.18	ketone	x
Methyl salicylate	119-36-8	152.15	ester	√
Hexanoic acid, methyl ester	106-70-7	130.18	ester	√
Butanoic acid, 2-methyl-, hexyl ester	10032-15-2	186.29	ester	x
Decane	124-18-5	142.28	alkane	√
Undecane, 3-methyl	1002-43-3	170.33	alkane	x
Dodecane	112-40-3	170.33	alkane	√
Tetradecane	629-59-4	198.39	alkane	√
Acetic acid	64-19-7	60.05	carboxylic acid	√
Hexanoic acid	142-62-1	116.16	carboxylic acid	√
Butanoic acid, 4-hydroxy	591-81-1	104.1	carboxylic acid	x
Furan, 2-pentyl	3777-69-3	138.21	furan	√
Caffeine	58-08-2	194.19	heterocyclic compound	x

**Table 3 foods-13-03996-t003:** Comparison of the two techniques GC-MS-FID and SIFT-MS (the symbol − means that this parameter is a disadvantage, +/− means that this parameter is neither an advantage nor a difficulty, the symbol ++ means that this parameter is an advantage).

	SPME-GC-FID-MS	SIFT-MS
**Analysis time**	− (35 min)	++ (3 min)
**Physicochemical selectivity**	Depends on SPME and GC column phases	Depends on reactivity of precursor ions
**Compound identification**	++	+/− (unit mass resolution)
**Quantification in complex matrices**	++ (possible with internal or external calibration)	− (conflict ions)
**Volatile profile discrimination**	++	++

## Data Availability

The original contributions presented in the study are included in the article/Appendix A, further inquiries can be directed to the corresponding author.

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
