# Peer review of "Comparison of Untargeted and Markers Analysis of Volatile Organic Compounds with SIFT-MS and SPME-GC-MS to Assess Tea Traceability"

_foods, 2024, doi:10.3390/foods13243996_

Round 1
Reviewer 1 Report
Comments and Suggestions for Authors
The article provides valuable insights into using different analytical techniques such as SIFT-MS and HS-SPME-GC-MS to identify and assess volatile organic compounds (VOCs) in tea samples for traceability purposes. The study effectively contrasts targeted marker-based analysis with untargeted profiling, highlighting each method’s capabilities and limitations. This work addresses a relevant topic in food traceability, focusing on a popular beverage with considerable economic and cultural importance. The authors show that while targeted HS-SPME-GC-MS successfully discriminated samples by origin using a set of forty-eight markers, it could not distinguish them based on color. On the other hand, untargeted approaches enabled better discrimination for both origin and color. This work is well-conceived, offering promising results for enhancing tea traceability and quality assessment using VOC profiles, though limited by the small sample size. The discussion thoroughly examines each method's strengths and weaknesses, offering valuable insights for future research. However, expanding the sample size including more diverse tea types could strengthen the study’s implications for broader applications. The following suggestions may be considered:
Comments/Suggestions:
1-First, we would like to share the statistics of the tea-producing countries as quoted in introduction lines 42 and 43. Is Pakistan still a tea-producing country? Moreover, according to the news, Sri Lanka is fourth among tea-producing countries, why is absent from the largely produced list in line 43?
2-As we move toward Table 1, it is noted that the sequence shows some randomness. It would be better and more convenient if it is arranged by countries(origin) and their sample sizes in the left column, not by color. It would be even better if you arranged it alphabetically.
3-The sample selection criteria surprisingly excluded some origins that have long been globally recognized for their unique aroma, quality, and composition, including Kenya, which ranks among the top three in production. Including it would be a good omen and lead to more reliable, reproducible outcomes, enhancing the authenticity of the result.
4-The limited number of tea samples used in the study restricts the generalizability of the results. Expanding the sample size could improve the robustness of the conclusions. Including missing samples in this work as mentioned and highlighted in the above comment would enhance the reproducibility of the instruments.
5-While they mentioned the selection of markers in 175-176 lines, an additional explanation of how the forty-eight markers were chosen and why they may have failed to differentiate by color would add clarity.
6-The last but not least, providing more details on the data processing and statistical methods used, especially in the untargeted approach, would help readers understand the analytical rigor and reproducibility of the results. The methods used are effective but overly complex and convoluted, making them less accessible, particularly for beginners or those less experienced in the field. The presentation would be more appealing and user-friendly if the steps were simplified, sequentially organized, and formatted with numbers or bullet points—especially in sections 2.2, 2.3, and 2.4 of the materials and methods section.
Author Response
Comments 1: First, we would like to share the statistics of the tea-producing countries as
quoted in introduction lines 42 and 43. Is Pakistan still a tea-producing country? Moreover,
according to the news, Sri Lanka is fourth among tea-producing countries, why is absent
from the largely produced list in line 43?
Response 1: We thank the reviewer for her/his nice advices and we've taken up the
introduction and added a few information on tea market and tea-exporting countries. Pakistan
is indeed a tea-producing country and is the 7th largest country in terms of black tea
consumption per capita. Tea Chaï is a famous black tea that would be very interesting to
include in our collection. Unfortunately, our work was based on samples directly picked up
from producers by our partner Lydia Gautier as a tea expert, without any contribution of
commercial firm. This aspect allowed us to study origin of samples without worrying about
the effects of mixing. Actually, it would be extremely interesting to complete this study with
a larger number of samples, from more countries, but on condition that their traceability is
well guaranteed. We therefore believe that our work could be taken up by the sector's supply
chains and companies.
Comments 2: As we move toward Table 1, it is noted that the sequence shows some
randomness. It would be better and more convenient if it is arranged by countries(origin)
and their sample sizes in the left column, not by color. It would be even better if you
arranged it alphabetically.
Response 2: Thank you for pointing this out. We modified the table 1 accordingly
Comment 3: The sample selection criteria surprisingly excluded some origins that have
long been globally recognized for their unique aroma, quality, and composition, including
Kenya, which ranks among the top three in production. Including it would be a good omen
and lead to more reliable, reproducible outcomes, enhancing the authenticity of the result.
Response 3: We agree with this comment. Therefore, the absence of teas from Kenya is
simply due to the structuring of the industry itself. As presented by Kagira et al 2011, Tea
growing in Kenya has two separated sectors: large scale sectors with multinationals and
small-scale sector with growers who sell their tea through small holder factories with several
mixing steps. In addition, our collaborator does not have a sufficient network to source
samples of Kenyan tea.
Comment 4: The limited number of tea samples used in the study restricts the
generalizability of the results. Expanding the sample size could improve the robustness of
the conclusions. Including missing samples in this work as mentioned and highlighted in
the above comment would enhance the reproducibility of the instruments.
Response 4: We agree with reviewer 1 that a larger sample collection would reinforce the
impact of this article and we hope that this work will be adopted by the tea sector.
Comment 5: While they mentioned the selection of markers in 175-176 lines, an additional
explanation of how the forty-eight markers were chosen and why they may have failed to
differentiate by color would add clarity.
Response 5: Thank you for pointing this out. We reinforce the arguments that guided our
choice of these 48 markers in the text Lines 198-203: From the set of chromatograms recorded
with this tea collection, we looked for compounds that met the following requirements: they
had to be well separated from other constituents by the chromatographic method, have a high
confidence score compared to NIST data (>70%) for mass spectrometry identification, and
present a strong FID signal. We thus selected 48 main compounds expressed in the
headspaces of our samples including.
Comment 6: The last but not least, providing more details on the data processing and
statistical methods used, especially in the untargeted approach, would help readers
understand the analytical rigor and reproducibility of the results. The methods used are
effective but overly complex and convoluted, making them less accessible, particularly
for beginners or those less experienced in the field. The presentation would be more
appealing and user-friendly if the steps were simplified, sequentially organized, and
formatted with numbers or bullet points—especially in sections 2.2, 2.3, and 2.4 of the
materials and methods section.
Response 6: The section 2.4 on data treatment was modified to clarify the processing. For
untargeted approaches we based our work on previous studies: Spadafora et al 2022 and
Reyrolle et al 2022. These articles were dedicated to the data treatment strategies and details.
Reviewer 2 Report
Comments and Suggestions for Authors
After reviewing the manuscript "Comparison of untargeted and markers analysis of volatile organic compounds with SIFT-MS and SPME-GC-MS to assess tea traceability", I have the following comments:
1. The abstract does not adequately explain the significance of using both techniques (SIFT-MS and SPME-GC-MS) or justify why these were chosen for tea traceability.
2. In the SIFT-MS method section, it would be helpful to provide a reason for selecting the specific positive and negative ion sources.
3. Graphs in the PLS-DA analysis are challenging to interpret without clearer grouping explanations. Using more contrasting symbols for each sample group could enhance readability.
4. Sparse PLS-DA analysis was used, but the criteria for selecting components (PC1, PC2, etc.) in visual representation are unclear. Additionally, the threshold for separating groups in PCA/PLS-DA is not explained in detail.
5. Inconsistent terminology: Terms like "volatile profiles," "VOCs," and "markers" are used interchangeably without clear definitions, which can confuse readers.
6. Additional figure showing some of the obtained mass spectrum would bring clarity in result interpretation.
Author Response
Comments 1: The abstract does not adequately explain the significance of using both techniques (SIFT-MS and SPME-GC-MS) or justify why these were chosen for tea traceability.
Response 1: Thank you for pointing this out. The abstract has been revised to enhance the aims of this work.
Comments 2: In the SIFT-MS method section, it would be helpful to provide a reason for selecting the specific positive and negative ion sources.
Response 2: SIFT-MS technic is a chemical ionization mass spectrometry working with up to 8 different precursor ions (3 positives and 5 negatives). Working with both polarity ions increases the selectivity and the possibility to detect different nature of molecules.
Comments 3: Graphs in the PLS-DA analysis are challenging to interpret without clearer grouping explanations. Using more contrasting symbols for each sample group could enhance readability.
Response 3: Thank you for pointing this out. The graphs have been revised accordingly. We expect that the addition of colors would enhance the readability of the graphs
Comment 4: Sparse PLS-DA analysis was used, but the criteria for selecting components (PC1, PC2, etc.) in visual representation are unclear. Additionally, the threshold for separating groups in PCA/PLS-DA is not explained in detail.
Response 4: MixoMics is a tool designed for the analysis of complex multivariate data, often in the field of metabolomics or microbiome. When performing an sPLS-DA analysis with MixoMics, the software follows a cross-validation process to determine the appropriate threshold for variable selection. For us, considering the difficulty of obtaining perfectly traced samples, it seemed sufficient to show that sample groupings could be obtained qualitatively. Quantitative results with predictive threshold require, in our view, a very large number of samples, which would be difficult to collect without the participation of tea-producing company.
Comments 5: Inconsistent terminology: Terms like "volatile profiles," "VOCs," and "markers" are used interchangeably without clear definitions, which can confuse readers.
Response 5: Thank you for pointing this out. A sentence has been added in section 2.4 to clarify the used terms.
Comments 6: Additional figure showing some of the obtained mass spectrum would bring clarity in result interpretation.
Response 6: supplementary figures were added to illustrate the HS-SPME-GC-MS analysis and the SIFT-MS fingerprints.
Reviewer 3 Report
Comments and Suggestions for Authors
The work may be of moderate interest as a comparison of analytical methods for a food application, however, the study is limited to the illustration of qualitative results on a small number of samples.
I would suggest improvements in post-processing and in the description of results, especially those concerning data processing.
There are some improvements or errors to be corrected:
· The description of the input data to the sPLS-DA algorithm must be improved with a more complete and clear description of the encodings and pre-treatments that may have been undergone by the input data. The exact identification of the data matrices submitted to sPLS-DA in the different combinations (targeted, untargeted) with instrumental methods is also lacking.
· Sparse PLS-DA, a classification algorithm, is used, however, the authors do not discuss any quantitative parameters regarding the goodness of the classification (specificity, accuracy, precision, ...) and the prediction error (e.g. validation method); is it because of the low number of items in the classes?.
· The results of the algorithm are described only with regard to the qualitative aspect connected with grouping on the projections of the latent variables as if it were a PCA algorithm, so why not use PCA?.
· The innovation brought by the present study to current knowledge should be highlighted in the conclusion section.
· I have not found the supplementary material referred to in the text (“(Table 1 SI)” in line 358).
· Lines 111 and 134 change "mL min-1" to "mL min-1"
· Lines 151-157 the description of the treatments presented here should be improved with the indication or description of the algorithms applied with the related parameters
· Lines 163-170 sentences are extraneous to the context of the work
· Line 195 "50 different markers" Previously, in line 175 and table 2, only 48 molecules were identified as markers. Make the identification and counting of markers consistent and unambiguous all over the text.
· Sections 3.1.1 and 3.2.1 the processing is discussed with respect to provenance, however, in these paragraphs the grouping of samples is attributed to the different cultivars. The authors should better discuss this aspect, is there a correlation between origin and cultivar? To what extent? If so, is the cultivar the real discriminator?
· Lines 211, 246, 249, 261 indicate references to the bibliography with the correct syntax
· Line 259 "This work with markers was performed with the 50 identified molecules", also elsewhere in the text 50 molecules are indicated but, previously, in line 175 and table 2, only 48 molecules were identified. Make identification and counting consistent and unambiguous throughout the article.
· Lines 358-359 "only 28 of the 50 products" is a reference to the marker substances in Table 2? What does "products" mean?
· Lines 378-380 I would suggest rewriting the sentence “It seems clear from these results that the quantification of markers without separation results in a loss of performance and that a profile comparison is relevant.” to make it easier to understand.
· Line 389 in the legend of table 3 indicates the meaning of the symbols ++ +/- etc. used.
Author Response
Comments 1: The work may be of moderate interest as a comparison of analytical methods for a food application, however, the study is limited to the illustration of qualitative results on a small number of samples. I would suggest improvements in post-processing and in the description of results, especially those concerning data processing.
Response 1: We agree that the number of samples could be considered limited in our study. Nevertheless, the main objective of this paper was to highlight that post processing of the HS-SPME-GC-MS measurements already recorded would improve the conclusions. The second point was to demonstrate that untargeted processing of raw full scan SIFT-MS data could clearly cover the limitation with conflict ions. In our opinion these aspects are interesting.
There are some improvements or errors to be corrected:
Comments 2: The description of the input data to the sPLS-DA algorithm must be improved with a more complete and clear description of the encodings and pre-treatments that may have been undergone by the input data. The exact identification of the data matrices submitted to sPLS-DA in the different combinations (targeted, untargeted) with instrumental methods is also lacking.
Response 2: Thank you for pointing this out. To be honest, no data cleaning was carried out on the targeted GC-MS-FID analyses. In the interests of transparency, we have added the raw data table2 in SI. As far as non-targeted SIFT-MS analyses are concerned, a previous publication by our group details this information, and we didn't feel it relevant to repeat it here. Once again, data pre-processing remains very limited in any case (elimination of signals with variance equal to zero). Finally, as regards the non-targeted use of GC-MS chromatograms by the Chromcompare+ software package, the algorithms are licensed and are not available to us.
Comments 3: Sparse PLS-DA, a classification algorithm, is used, however, the authors do not discuss any quantitative parameters regarding the goodness of the classification (specificity, accuracy, precision, ...) and the prediction error (e.g. validation method); is it because of the low number of items in the classes?.
Response 3: The reviewer 3 perfectly identified our limitations. We only draw qualitative separations based on sPLS-DA analysis; The construction of predictive models would require larger collection of samples. But, such collection with perfectly traced samples is challenging for academic work without exposure to conflict of interest regarding tea-producing company.
Comments 4: The results of the algorithm are described only with regard to the qualitative aspect connected with grouping on the projections of the latent variables as if it were a PCA algorithm, so why not use PCA?.
Response 4: PCA is a descriptive method that does not allow spontaneous classification of our samples according to origin or color. Only cultivar seems to drive the spontaneous classification of samples on the basis of VOC emissions. The use of supervised analysis makes it possible to extract the relevant information for obtaining these classifications.
Comments 5: The innovation brought by the present study to current knowledge should be highlighted in the conclusion section.
Response 5: Thank you for pointing this out. We modified the conclusion accordingly with this sentence: “We do hope that this work will encourage teams who carry out traditional VOC analysis using marker lists to consider taking their raw data and analyzing it with non-targeted tools.”
Comments 6: I have not found the supplementary material referred to in the text (“(Table 1 SI)” in line 358).
Response 6: Thank you for pointing this out. We apologize for this oversight. Table 1 has been added in SII/We agree with this comment.
Comments 7: Lines 111 and 134 change "mL min-1" to "mL min-1"
Response 7: done
Comments 8: Lines 151-157 the description of the treatments presented here should be improved with the indication or description of the algorithms applied with the related parameters
Response 8: We apologize for not being able to provide more information on the algorithms included in the Chromcompare software. We have extracted the 400 features obtained by this software and analyzed them by sPLS-DA without any further data pre-processing.
Comments 9: Lines 163-170 sentences are extraneous to the context of the work
Response 9: Again, we apologize for this mistake and we corrected it.
Comments 10: Line 195 "50 different markers" Previously, in line 175 and table 2, only 48 molecules were identified as markers. Make the identification and counting of markers consistent and unambiguous all over the text.
Response 10: Thank you for pointing this out. We corrected it in the whole manuscript: 48 compounds were used in this work.
Comments 11: Sections 3.1.1 and 3.2.1 the processing is discussed with respect to provenance, however, in these paragraphs the grouping of samples is attributed to the different cultivars. The authors should better discuss this aspect, is there a correlation between origin and cultivar? To what extent? If so, is the cultivar the real discriminator?
Response 11: Thank you for pointing this particular point. We observed with our sample collection that cultivar seems to be the most contributing factor to VOC profiles of tea. To confirm that observation based on tea expert knowledge, additional analysis on phylogenetic distances of Camelia sinensis plants would be needed. This point requires skills we don't have. We have initiated a collaboration with a team specialized in this field. However, the processing of tea leaves makes this analysis difficult. A sentence has been added in the end of the section 3.2.1
Comments 12: Lines 211, 246, 249, 261 indicate references to the bibliography with the correct syntax
Response 12: Whole the bibliography has been checked and corrected.
Comments 13: Line 259 "This work with markers was performed with the 50 identified molecules", also elsewhere in the text 50 molecules are indicated but, previously, in line 175 and table 2, only 48 molecules were identified. Make identification and counting consistent and unambiguous throughout the article.
Response 13: These errors have been modified
Comments 14: Lines 358-359 "only 28 of the 50 products" is a reference to the marker substances in Table 2? What does "products" mean?
Response 14: Thank you for pointing this out. This mistake corresponds to French translation error. The text has been modified with 28 of the 48 compounds…
Comments 15: Lines 378-380 I would suggest rewriting the sentence “It seems clear from these results that the quantification of markers without separation results in a loss of performance and that a profile comparison is relevant.” to make it easier to understand.
Response 15: Thank you for pointing this out. We modified this sentence with “It seems clear from these results that the choice of markers is a key element for samples classification based on descriptive variables. The use of a untargeted analysis strategy for GC-MS and SIFT-MS measurements avoids any marker selection bias and maximizes sample sorting.”
Comments 16: Line 389 in the legend of table 3 indicates the meaning of the symbols ++ +/- etc. used.
Response 16: We added a description in the legend “Performances were evaluated qualitatively as negative (-), moderate (+/-), positive (+) and significantly positive (++).”
Reviewer 4 Report
Comments and Suggestions for Authors
The Manuscript entitled "Comparison of untargeted and markers analysis of volatile organic compounds with SIFT-MS and SPME-GC-MS to assess tea traceability" describe the profile/fingerprint VOC analysis to discriminate the origin and transformation processing of tea samples by SIFT-MS and SPME-GC-MS. According to the authors, a good discrimination of origin and color with both instruments was achieved. I would like to highlight some points that should be improved:
1) It is not clear which types of tea were studied, as well as which plant organs are present in each tea. It appears that the authors are only researching the Camellia sinensis tea variety, but the title of the paper and the descriptions throughout the text need to be clear.
2) In the abstract, the authors describe that the methodologies discriminated the origin and color of the tea, but in the results and discussion they describe that in some cases it was not possible to discriminate by color. The writing of the paper is confusing in some parts.
3) Regarding the volatile chemical profile by HS-SPME/GC-MS, it is essential to calculate the retention indices (RI) and compare them with values reported in literature. Additionally, it is important to describe the similarity indices (ex., NIST, Wiley) for each volatile compound identified to ensure accurate identification. Show a detailed table containing all the compounds identified by this methodology. The methodology, results and discussion need to be improved.
4) Section 2.4: Describe in detail which parameters were used for the statistical analyses, in addition to those described. For example: ion, compound name, percentage area, retention time. Are they necessary for statistical analysis?
5) It is also recommended to illustrate the chemical structures of the main volatile compounds identified in the study.
Minor issues
Line 188-192: Describe the meaning of the characters described in the last column.
Line 218: Change “PC1 and PC2 plot for the SIFT-MS data” to “PC2 and PC3 plot for the SIFT-MS data”
Line 256-257: A) PC1 and PC2 plot for the GC-FID data or A) PC1 and PC2 plot for the GC-MS/FID data?
Line 338-342: Rewrite the caption. It is wrong.
Author Response
Comments 1: The Manuscript entitled "Comparison of untargeted and markers analysis of volatile organic compounds with SIFT-MS and SPME-GC-MS to assess tea traceability" describe the profile/fingerprint VOC analysis to discriminate the origin and transformation processing of tea samples by SIFT-MS and SPME-GC-MS. According to the authors, a good discrimination of origin and color with both instruments was achieved. I would like to highlight some points that should be improved: It is not clear which types of tea were studied, as well as which plant organs are present in each tea. It appears that the authors are only researching the Camellia sinensis tea variety, but the title of the paper and the descriptions throughout the text need to be clear.
Response 1: We thank the reviewer for her/his nice advice. This study focused on tea samples meaning dried leaves of the Camellia sinensis. The leaves may be collected at different state of development (buds, young leaves or mature leaves) depending on country and label. The different types of tea was developed in the introduction that has been revised.
Comments 2: In the abstract, the authors describe that the methodologies discriminated the origin and color of the tea, but in the results and discussion they describe that in some cases it was not possible to discriminate by color. The writing of the paper is confusing in some parts.
Response 2: Thank you for pointing this out. We agree with this comment and we explained the limitation of the threshold with the small number of samples of each category. Therefore, we have added a sentence in the conclusion Line472-473: Separation according to origin and/or color could be greatly improved with a sample collection more suited to the purpose.
Comments 3: Regarding the volatile chemical profile by HS-SPME/GC-MS, it is essential to calculate the retention indices (RI) and compare them with values reported in literature. Additionally, it is important to describe the similarity indices (ex., NIST, Wiley) for each volatile compound identified to ensure accurate identification. Show a detailed table containing all the compounds identified by this methodology. The methodology, results and discussion need to be improved.
Response 3: We agree with this comment on retention Index. Nevertheless, the main aim of this paper was to illustrate the possibility to re interpret results already recorded with an unsupervised strategy. The HS-SPME-GC-MS-FID were collected without calibration of retention index and it was not be possible to perform it after.
Comments 4: Section 2.4: Describe in detail which parameters were used for the statistical analyses, in addition to those described. For example: ion, compound name, percentage area, retention time. Are they necessary for statistical analysis?
Response 4: We have updated section 2.4 to meet the reviewer’s expectations.
Comments 5: It is also recommended to illustrate the chemical structures of the main volatile compounds identified in the study.
Response 5: We feel that this would make the document more difficult to read, and we believe that the CAS numbers in table 2 are sufficient to answer this question.
Comments 6: Minor issues
Line 188-192: Describe the meaning of the characters described in the last column.
“the check marks indicate the known molecules, and the asterisks indicate the molecules that were excluded because of conflict ions.”
Line 218: Change “PC1 and PC2 plot for the SIFT-MS data” to “PC2 and PC3 plot for the SIFT-MS data”
done
Line 256-257: A) PC1 and PC2 plot for the GC-FID data or A) PC1 and PC2 plot for the GC-MS/FID data?
done
Line 338-342: Rewrite the caption. It is wrong.
Thank you for pointing this out. We modified the legend
Round 2
Reviewer 2 Report
Comments and Suggestions for Authors
After reviewing the revised manuscript, I have the following comments:
Overall, the revised manuscript appears to address the comments effectively. However, following issues remain.
1. The supplementary information is functional and provides critical details, but it could be improved in clarity, connection to the main text, and user-friendliness.
2. Enhancing visual aids with annotations and offering summaries of key data points would make it significantly more impactful.
3. The sheer size of the table SI1 makes it difficult to discern key insights. Also the table's readability need enhancements To improve:
- Highlight the most discriminative VOCs used in sPLS-DA analysis.
- Use bold text, color coding, or symbols to indicate compounds of high significance in differentiating tea origins or processing methods.
4. The units and significance of the values provided are unclear. For instance:
- Are these absolute intensities, normalized values, or semi-quantitative measurements?
- Clarify this in a footnote or a supplementary section to avoid confusion.
Author Response
Comments 1: The supplementary information is functional and provides critical details, but it could be improved in clarity, connection to the main text, and user-friendliness.
Reponse 1: We thank the rewiever for her/his nice advices to enhance the global quality of our paper. Several links to the SI were added in the main text (Line 142, Line 201 and Line 365). SI was also re-formated.
Comments 2: Enhancing visual aids with annotations and offering summaries of key data points would make it significantly more impactful.
Response 2: We apologize, but we do not see what improvements the reviewer is referring to.
Comments 3. The sheer size of the table SI1 makes it difficult to discern key insights. Also the table's readability need enhancements To improve: Highlight the most discriminative VOCs used in sPLS-DA analysis. Use bold text, color coding, or symbols to indicate compounds of high significance in differentiating tea origins or processing methods.
Response 3: The table S1 was updated and figure S3 SI was added to illustrate the most relevant features of sPLS-DA analysis of GC-MS features according to the process. Figure S4 was also added to illustrate the intesnities of several highlighted features.
Comments 4: The units and significance of the values provided are unclear. For instance: Are these absolute intensities, normalized values, or semi-quantitative measurements? Clarify this in a footnote or a supplementary section to avoid confusion.
Response 4: the GC-FID values were indicated as "pseudo-quantification" in line 166 and SIFT-MS values in multi ion monitoring were indicated in ppbV (Line 176). Details were recalled in the SI part.
Reviewer 4 Report
Comments and Suggestions for Authors
The authors have improved the manuscript and attached supplementary material. They addressed the critical point regarding the use of the retention index for compound identification. However, I suggest increasing the match percentage with the library to at least 85%, if possible. This adjustment will enhance the accuracy of the untargeted analysis in identifying the VOCs.
Author Response
Comments : The authors have improved the manuscript and attached supplementary material. They addressed the critical point regarding the use of the retention index for compound identification. However, I suggest increasing the match percentage with the library to at least 85%, if possible. This adjustment will enhance the accuracy of the untargeted analysis in identifying the VOCs.
response: We are grateful to the reviewer for his comments and take note of his advice to improve the efficiency of the unsupervised analysis. We have followed the advices of Markes support regarding features selection threshold settings.
Round 3
Reviewer 2 Report
Comments and Suggestions for Authors
Overall, the responses address most of the feedback constructively, with tangible improvements in linking the supplementary data to the main text, enhancing table clarity, and clarifying measurement units.
Comment 2: The visual aids presented in the manuscript are informative but could benefit significantly from enhancements to improve their interpretability and impact. Specifically:
- Annotations: Adding annotations to figures (e.g., marking discriminative VOCs or key trends with arrows or labels) would help readers quickly identify critical insights.
- Summarization: Including a concise summary table that highlights the most significant data points (e.g., top discriminative VOCs and their intensities or relevance to specific outcomes) would make the supplementary information more user-friendly.
- Visual Representation: Consider using bold text, color coding, or symbols to draw attention to compounds of particular importance in tables and graphs.
These changes would enhance the clarity and engagement of the visual materials, making the findings more accessible to a broader audience.
Author Response
Comment 1: Overall, the responses address most of the feedback constructively, with tangible improvements in linking the supplementary data to the main text, enhancing table clarity, and clarifying measurement units.
Response 1: We would like to thank reviewer 1 for his valuable advice, which helped us to improve the overall quality of our article.
Comment 2: The visual aids presented in the manuscript are informative but could benefit significantly from enhancements to improve their interpretability and impact. Specifically:
- Annotations: Adding annotations to figures (e.g., marking discriminative VOCs or key trends with arrows or labels) would help readers quickly identify critical insights.
- Summarization: Including a concise summary table that highlights the most significant data points (e.g., top discriminative VOCs and their intensities or relevance to specific outcomes) would make the supplementary information more user-friendly.
- Visual Representation: Consider using bold text, color coding, or symbols to draw attention to compounds of particular importance in tables and graphs.
These changes would enhance the clarity and engagement of the visual materials, making the findings more accessible to a broader audience.
Response 2: A new table has been added in the supplementary information reporting the contribution of each molecule to discrimination according to the origin (line249-250): “The contributions of the 48 molecules to the classification of the samples according to their origin are shown in Table 1 SI.” In addition, pictograms were also added in the test (Lines 235 to 244 and 319 to 332) to highlight the links between results and graphs.
Reviewer 4 Report
Comments and Suggestions for Authors
The authors have made the suggested changes in the previous revisions. I recommend the manuscript for publication.
Author Response
Comment 1: The authors have made the suggested changes in the previous revisions. I recommend the manuscript for publication.
Response 1: We would like to thank reviewer for his valuable advices, which helped us to improve the overall quality of our article.